# Docking Studies, Cytotoxicity Evaluation and Interactions of Binuclear Copper(II) Complexes with S-Isoalkyl Derivatives of Thiosalicylic Acid with Some Relevant Biomolecules

**DOI:** 10.3390/ijms241512504

**Published:** 2023-08-06

**Authors:** Jelena D. Dimitrijević, Natalija Solovjova, Andriana M. Bukonjić, Dušan Lj. Tomović, Mirjana Milinkovic, Angelina Caković, Jovana Bogojeski, Zoran R. Ratković, Goran V. Janjić, Aleksandra A. Rakić, Nebojsa N. Arsenijevic, Marija Z. Milovanovic, Jelena Z. Milovanovic, Gordana P. Radić, Verica V. Jevtić

**Affiliations:** 1Center for Harm Reduction of Biological and Chemical Hazards, Faculty of Medical Sciences, University of Kragujevac, Serbia, Svetozara Markovića 69, 34000 Kragujevac, Serbia; jelasavka@hotmail.rs (J.D.D.); mirjanamilinkovic4@gmail.com (M.M.); marijaposta@gmail.com (M.Z.M.); jelenamilovanovic205@gmail.com (J.Z.M.); 2Academy of Applied Studies Belgrade, The College of Health Science, Cara Dušana 254, 11080 Belgrade, Serbia; natalijas-63@yandex.ru; 3Department of Pharmacy, Faculty of Medical Sciences, University of Kragujevac, Svetozara Markovica 69, 34000 Kragujevac, Serbia; andriana.bukonjic@hotmail.com (A.M.B.); dusantomovic88@hotmail.com (D.L.T.); 4Department of Chemistry, Faculty of Science, University of Kragujevac, Radoja Domanovic 12, 34000 Kragujevac, Serbia; angelina.cakovic@pmf.kg.ac.rs (A.C.); jovana.bogojeski@pmf.kg.ac.rs (J.B.); wor@kg.ac.rs (Z.R.R.); 5National Institute of the Republic of Serbia, Department of Chemistry, Technology and Metallurgy, University of Belgrade-Institute of Chemistry, Njegoševa 12, 11000 Belgrade, Serbia; goran.janjic@ihtm.bg.ac.rs; 6Faculty of Physical Chemistry, University of Belgrade, Studentski trg 12-16, 11158 Belgrade, Serbia; saska@ffh.bg.ac.rs; 7Faculty of Medical Sciences, Department of Microbiology and Immunology, University of Kragujevac, Svetozara Markovića 69, 34000 Kragujevac, Serbia; arne@medf.kg.ac.rs; 8Center for Molecular Medicine and Stem Cell Research, Faculty of Medical Sciences, University of Kragujevac, Svetozara Markovića 69, 34000 Kragujevac, Serbia; 9Faculty of Medical Sciences, Department of Histology and Embryology, University of Kragujevac, Svetozara Markovića 69, 34000 Kragujevac, Serbia

**Keywords:** binuclear copper(II) complexes, DNA interaction, docking studies, colon cancer, CT26 cells, cytotoxicity, antitumor activity, inflammation

## Abstract

The numerous side effects of platinum based chemotherapy has led to the design of new therapeutics with platinum replaced by another transition metal. Here, we investigated the interactions of previously reported copper(II) complexes containing S-isoalkyl derivatives, the salicylic acid with guanosine-5′-monophosphate and calf thymus DNA (CT-DNA) and their antitumor effects, in a colon carcinoma model. All three copper(II) complexes exhibited an affinity for binding to CT-DNA, but there was no indication of intercalation or the displacement of ethidium bromide. Molecular docking studies revealed a significant affinity of the complexes for binding to the minor groove of B-form DNA, which coincided with DNA elongation, and a higher affinity for binding to Z-form DNA, supporting the hypothesis that the complex binding to CT-DNA induces a local transition from B-form to Z-form DNA. These complexes show a moderate, but selective cytotoxic effect toward colon cancer cells in vitro. Binuclear complex of copper(II) with S-isoamyl derivative of thiosalicylic acid showed the highest cytotoxic effect, arrested tumor cells in the G2/M phase of the cell cycle, and significantly reduced the expression of inflammatory molecules pro-IL-1β, TNF-α, ICAM-1, and VCAM-1 in the tissue of primary heterotopic murine colon cancer, which was accompanied by a significantly reduced tumor growth and metastases in the lung and liver.

## 1. Introduction

Cancer is undoubtedly a potentially life-threatening disease for humanity that can greatly affect the quality of human life [1,2,3,4,5]. The numerous side effects of cisplatin such as nausea, kidney and liver failure (typical of heavy metal toxicity) have induced the search for better alternative candidates to non-platinum metal complexes such as copper, cobalt, nickel, zinc, ruthenium, and iron [6,7,8,9,10]. Copper is the third most abundant metal, behind iron and zinc, and is found in the body in trace amounts. The total amount of copper in the human body is only 75–100 mg, but is present in every tissue of the body. It is primarily stored in the liver, with smaller amounts found in the brain, heart, kidneys, and muscles [7]. As an essential trace element, copper plays a very important role in many physiological cellular processes [8,9]. Metal complexes of the Schiff base ligands (coordinated via ONS and NO donor atoms) with copper(II) ions are interesting due to their structural, spectral, and redox properties [10]. Complexes containing a sulfur atom can be expected to possess great potential for medical applications because such molecules may mimic the functional properties of sulfur-containing proteins. Sulfur is well-known to be an important component of biomolecules and it plays a key role in biological systems [11]. Cytotoxic activity is primarily mediated by targeting DNA and proteins. Interactions with small molecules leads to DNA damage in cancer cells by blocking uncontrolled division, resulting in cell death [12,13]. Many inorganic compounds have been successfully used as drugs in the treatment of different types of cancers. Copper(II) complexes have been considered and suggested to be better candidates for cancer treatment due to their significant role in biological systems, having negligible or no side effects [14,15]. Mononuclear copper(II) complexes with the phenanthroline type of ligand show lesser affinities to DNA binding compared to binuclear copper(II) complexes [16,17]. In angiogenesis, copper appears to be involved in stimulating proliferation and endothelial cell migration, and acts as a cofactor for angiogenic factors (VEGF, bFGF, TNF-α, and IL-1) [18,19,20,21]. However, in embryogenic cells, the human copper transporter (hCTR1) inhibits the activation of cellular signaling pathways, resulting in the development and progression of cancer [22]. The difference in the tumor cells’ response to copper compared to normal cells probably laid the basis for the development of anticancer drugs [15,23].

The aim of the present study was to research the interactions of copper(II)complexes with S-isoalkyl derivatives of thiosalicylic acid (isoalkyl = isopropyl-, isobutenyl-, and isopentyl) marked as **C1**–**C3** with guanosine-5′-monophosphate (5′-GMP) and calf thymus DNA (CT-DNA) as well as the evaluation of their in vitro and in vivo antitumor activity in a mouse colon cancer model. In this article, we present a docking study of binuclear copper(II) complexes with S-isoalkyl derivatives of thiosalicylic acid. Previously, we showed the synthesis, characterization, and cytotoxic effect of newly synthetized copper(II) complexes (**C1**–**C3**) with isoalkyl derivatives of thiosalicylic acid as ligands (**L1**–**L3**). Their chemical structures are presented in Figure 1. These complexes showed a lower cytotoxic effect than cisplatin toward normal fibroblasts as evaluated by the MTT assay as well as significant apoptotic effects against HCT116 cells. We also found antiproliferative effects of **C1**, a copper(II)complex with an isopropyl derivative of thiosalicylic acid toward HCT116 cells [24]. All complexes induced the arrest of HCT116 cells in the G2 phase of the cell cycle [24]. Taking into account the well-known anti-inflammatory activity of derivatives of salicylates [25] and the involvement of inflammatory pathways in cancerogenesis and the progression of cancer [26], the anticancer properties of transition metal complexes with different derivatives of salicylates could be considered to have higher in vivo effects in comparison with their effects in cell culture systems. By reducing inflammation, salicylates are known to reduce the risk of large bowel cancer development [27]. Hence, in this study, we analyzed the antitumor effects of copper(II) complexes with isoalkyl derivatives of thiosalicylic acid against murine colon carcinoma cells, CT26, and we also explored the in vivo antitumor effect of the most active complex in a murine heterotopic model of colon carcinoma.

## 2. Results

### 2.1. DNA Interactions Studies

#### 2.1.1. UV–Vis Absorption Studies

Electronic absorption spectroscopy is widely used to determine the DNA binding affinity of the complexes to CT-DNA. The typical titration curves for **C1**–**C3** in the presence of CT-DNA at different concentrations are given in Figure 1. The addition of CT-DNA to the complexes **C1**–**C3** showed a hyperchromism (Figure 1). This hyperchromism shift indicated the interaction of the CT-DNA with the complexes. In general, the extent of the hypochromism or hyperchromism observed provides a measure of the strength of binding of the complexes to CT-DNA [28,29]. Among the studied Cu(II) complexes, **C2** showed a significant binding affinity toward CT-DNA (Table 1). The equilibrium binding constant (*K*_b_) between CT-DNA and each of the complexes was calculated using Equation (1). Moreover, the intrinsic binding constant *K*_b_ (Table 1) obtained for the studied complexes with CT-DNA followed the order **C2** > **C1** > **C3**, indicating that **C2** interacted more strongly than the other complexes.

#### 2.1.2. Ethidium Bromide (EB) Displacement Studies

Ethidium bromide (EB) emits an intense fluorescence light in the presence of DNA due to strong intercalation between the adjacent DNA base pairs. The enhanced fluorescence can be quenched by the addition of a second molecule [30,31]. The competitive binding experiments were carried out on EB–CT-DNA by varying the concentration of the Cu(II) complexes. The extent of the fluorescence of EB bound to DNA is used to determine the extent of binding between the second molecule and DNA [32]. The interactions of **C1**–**C3** with CT-DNA were studied with an EB-bound CT-DNA solution in PBS (pH = 7.2). The interaction of **C1**–**C3** with CT-DNA led to a quenching in fluorescence intensity due to the displacement of EB from the EB–DNA complex. The quenching parameters for **C1**–**C3** were calculated using the Stern–Volmer equation. EB displacement studies were performed by changing the concentration of the metal complexes and monitoring the emission intensity of the EB–DNA complex [33,34]. An increase in the concentration of **C1**–**C3** (0–60 μM) exhibited a significant decrease in fluorescence intensity with a red shift (Figure 2). This indicates that EB was released from the EB–DNA complex due to its replacement by the studied complexes. Fluorescence quenching data were analyzed using the Stern–Volmer equation (Equation (2)) and a quenching constant (*K*_sv_) was calculated from I_0_/I vs. the [Q] of **C1**–**C3**, and the obtained values are presented in Table 1.

Fluorescence quenching data could also be used to determine the binding sites (*n*) and the equilibrium binding constant *K*_bin_ by using the Scatchard equation [35,36]:log(I_0_ − I)/I = log*K*_bin_ + *n*log[Q](1)

The number of binding sites (*n*) and the binding constant (*K*_bin_) were calculated from the plot of log(I_0_ − I)/I vs. log[Q] (Table 1). Based on the values of *K*_sv_ and *K*_bin_ given in Table 1, it can be noted that there was an interaction between the used Cu(II) complexes and CT-DNA. The complex **C2** has a greater affinity to interact with CT-DNA than the other complexes. However, the addition of the complexes to the EB bound CT-DNA did not quench (2.5% for **C1**, 8% for **C2** and 7% for **C3**) the emission intensity of the EB bound CT-DNA around 610 nm significantly (Figure 2). The obtained little quenching along with small values of *K*_sv_ clearly show that the Cu(II) complexes did not interact with the CT-DNA by intercalation.

#### 2.1.3. Viscosity

For further establishment of the interactions between the complexes and DNA, viscosity measurements were carried out. A classical intercalation model demands that the DNA helix must lengthen as base pairs are separated to accommodate the binding ligand, which leads to the increase in DNA viscosity. Ethidium bromide, a well-known DNA intercalator, strongly increases the relative viscosity by lengthening the DNA double helix through intercalation. In order to further confirm the modes of binding of the complexes, viscosity measurements of DNA solutions were performed in the presence and absence of these complexes. In classical intercalation, the complexes result in lengthening and stiffening of the double helix of DNA, leading to an increase in the viscosity of DNA [37,38]. The addition of increasing amounts (up to r = 1.0) of **C1**–**C3** to a CT-DNA solution (0.01 mM) resulted in an increase in the relative viscosity of CT-DNA (Figure 3), which was more pronounced upon the addition of **C2** complexes, while **C1** and **C3** showed a moderate increase in the relative viscosity of CT-DNA. Therefore, the observed result suggests that the complexes could interact with CT-DNA (UV–Vis measurements) but it is unlikely that they interact by intercalation (viscosity and ethidium bromide (EB) displacement studies). The exception may be complex **C2** if we consider the viscosity results; however, the obtained *K*_sv_ constant still had a low value.

#### 2.1.4. Protein Binding Studies

A possible way of the biotransformation and the mechanism of action of the chemotherapeutic agents (among which are metal drugs) could be the interaction between small molecules and serum protein [39]. Bovine serum albumin (BSA) is often selected as a model protein to study the interaction of the small molecules with serum albumin due to its similarity with human serum albumin [40]. Therefore, besides the study of interactions of Cu(II) complexes toward CT-DNA, further investigation of the interaction of complexes **C1**–**C3** with bovine serum albumin (BSA) was undertaken. Qualitative analysis of the interaction of complexes **C1**–**C3** with BSA was studied by examining the quenching of the tryptophan fluorescence emission spectra of BSA in the presence of the complexes. The fluorescence of the protein is specifically caused by three amino acid residues, namely tryptophan, tyrosine, and phenylalanine. BSA consists of two tryptophan residues along its amino acid sequence and exhibits tryptophan fluorescence at an excitation of 295 nm with an emission maximum at 365 nm.

In the present study, we examined the affinity of **C1**–**C3** for BSA by using tryptophan fluorescence quenching experiments. Fluorescence spectroscopy can monitor changes in the protein structure, dynamics, and folding [41,42,43]. The change in BSA fluorescence upon the addition of increasing concentrations of **C1**–**C3** (0–20 µM) over the range 300–500 nm (λ_ex_, = 295 nm) is presented in Figure 4. As shown, a decrease in fluorescence intensity at 362, 363, or 364 nm was observed. Fluorescence quenching data were analyzed using the Stern–Volmer equation (Equation (3)) and a quenching constant (*K*_sv_) was calculated from I_0_/I vs. [Q] (Figure 4).

The quenching constants for **C2** and **C3** were found to be similar (Table 2), and the quenching ability of **C2** and **C3** was two times greater than the **C1** complex. The quenching constant of the complexes was in the order of **C1** > **C3** > **C2** (Table 2), which was similar to the order of the binding constant of the complexes with CT-DNA, as observed earlier (Table 1). The extent of quenching of the fluorescence intensity assigns a measure of association of the complex with BSA. Small molecules bind independently to a set of sites in a protein molecule, and the equilibrium between the free and the bound molecule is given by the Scatchard equation [35,36], Equation (2).

The binding constant (*K*_bin_) and the number of binding sites (n) were obtained from the plot of log(I_0_ − I)/I vs. log[Q] (Table 2). However, complex **C2** showed 10^6^ fold more binding affinity than the **C1** complex, whereas the **C1** complex showed 10^3^ fold more binding affinity than **C3**, as depicted from their binding constants *K*_bin_ (Table 2). The binding constant of the complexes toward BSA also followed the same order as observed for the quenching constant (Table 2). The binding stoichiometry of BSA with Cu(II) complexes was also found to be greater for the **C2** complex than the other complexes (Table 2).

From the results of the interaction of the complex with the DNA and BSA molecules, it appears that in both cases, the **C2** complex had the best binding affinity to both molecules. The difference in the structure of the complex was only in the side chain of the isopropyl, isobutenyl, and isopentyl groups, while the **C2** complex has an isobutenyl group in the side chain. The binding affinity is equally affected by steric and electronic effects. It can be concluded that **C2** had the highest affinity because it has a longer side chain, so it can better penetrate the DNA and BSA molecules, but is not too long, like **C3**, to cause steric hindrance.

### 2.2. Results of Docking

The docking study aims to cover two distinct types of DNA structures. One type is a canonical double-strand Watson–Crick structure containing no intercalation site, known as B-form DNA (B-DNA, PDB code: 1BNA). The second target involves one of the non-canonical DNA that exhibits a B-form structure in one region, while the other half adopts a non-canonical Z-form structure (Z-DNA, PDB code: 2ACJ).

The results of molecular docking on the B-form of DNA revealed that all three complexes exhibited a slightly higher affinity for binding to the minor groove than to the major groove. The binding energies for the minor groove (BS1, Figure 5) were −7.2 kcal/mol (for **C1** complex), −8.1 kcal/mol (for **C2** complex), and −8.2 kcal/mol (for **C3** complex). The fluorescence quenching study revealed the presence of two binding sites (n) exclusively for the **C2** complex, whereas the other two complexes each occupied only one binding site. The second binding site for the **C2** complex was located in the major groove, displaying a slightly lower binding energy (−7.9 kcal/mol) compared to the minor groove.

Supramolecular analysis of all three investigated Cu(II) complexes (**C1**, **C2**, and **C3**) at the first binding site on the B-DNA demonstrated the participation of three thiosalicylic bridging ligands of the complexes in binding to the minor groove. However, this binding mode does not conform to the classical minor groove interaction, as the complexes interact simultaneously with nucleobases, sugar units, and the phosphate backbone. Among these interactions, those with the phosphate backbone are the most numerous. Since all three complexes are bound at the same site in the minor groove and exhibit considerable structural overlap, only the interactions of the **C2** complex are shown in Figure 6 for convenient comparison with its binding in the major groove. Upon binding to the major groove, all four thiosalicylic bridging ligands of **C2** are involved in interactions with the DNA. In this case, interactions with nucleobases are slightly more numerous than interactions with the phosphate backbone (Figure 6).

Previous experiments have demonstrated that the addition of increasing amounts of **C1**–**C3** to a solution of circulating tumor DNA (CT-DNA) leads to an elevation in the relative viscosity of the CT-DNA solution (Figure 3). Furthermore, the introduction of complexes into the solution of ethidium bromide (EB)-bound CT-DNA was slightly quenched (shown a slight reduction) in EB emission (Figure 2). These observations suggest that the complexes do not interact with CT-DNA through intercalation, and that the increase in viscosity is a consequence of DNA elongation resulting from a structural change, specifically, the transition from the B-form to the Z-form of DNA. It is evident that the binding of the complexes in the minor groove (BS1) induces DNA elongation, likely due to their interactions, besides nucleobases as well as with sugar units and phosphate backbone. On the other hand, the binding of **C2** in the major groove causes more significant changes in the DNA structure, leading to pronounced DNA elongation. This can be attributed to a higher number of simultaneous interactions with nucleobases and the phosphate backbone as well as a smaller steric effect and the easier access of **C2** to the nucleobases.

B-form DNA is commonly found in living organisms under physiological conditions, characterized by high hydration and low salinity. A canonical DNA structure adopts a right-handed double helix structure, with base pairs in an anti-conformation, forming minor and major grooves. The flexibility of the helix allows B-DNA to adopt several non-canonical structural forms. On the other hand, the left-handed helicity and elongated double-stranded structure are unique to the higher-order Z-DNA. Z-DNA is named after the zig-zag conformation of the sugar–phosphate backbone, which arises due to alternating syn-anti conformations of the base pairs. This base pair conformational pattern causes increased intrastrand phosphate–phosphate distances and the elongation of the Z-DNA structure compared to all other forms of DNA molecules [44]. The anti-conformation of nucleobases is energetically more favorable due to fewer steric clashes. However, purine basis can more easily adopt syn conformations than pyrimidine bases. The interconversion of purine bases from the anti to the syn state is easier since these states are almost isoenergetic. Furthermore, guanine is more likely to adopt syn conformation than thymine. The syn conformation of guanine is stabilized by the crystal water-mediated hydrogen bond networks of guanine that involve phosphate groups. Non-APP sequences may convert to Z-conformation under more severe conditions (e.g., high salt concentration) [45]. Z-DNA is typically formed in regions with alternating purine–pyrimidine dinucleotide repeat sequences, commonly observed in the guanine–cytosine (G–C) rich areas. The propensity of B-DNA to convert to Z-DNA is lower in the cytosine–adenine (C–A) rich areas, while the least likely conformation change occurs in the adenine–thymine (A–T) rich regions of DNA [46]. The bases are pulled more toward the outside of the helix, which makes the Z-DNA structure more rigid and compact than B-DNA. The increased rigidity reduces the entropy change, and the enthalpy of transition into Z-DNA is positive. Another consequence of the increase in structural rigidity is the water release. The Z-DNA conformation is entropy-driven, since an overall entropy gain is a consequence of double-strand dehydration. Dehydration agents such as salts, zwitterions, monohydroxyl alcohols, and polyols as well as osmolytes such as amino acids, sugars, methylamines, methylsulfonium compounds, and urea promote the transition into the Z-form [45].

The formation of Z-DNA can be induced under various circumstances that provide a positive-charge environment for the stabilization of the negatively-charged Z-DNA surface. These circumstances include a high ionic environment, small molecules, metallic-complexes, alcohols, negative supercoiling of DNA, base-modification, Z-DNA binding proteins, various proteins, and small basic peptides. The Z-form of DNA plays important roles in essential cellular events such as transcriptional regulation, immune response, and double-strand DNA breaks. In order for a nucleotide sequence in the B-form to adopt the Z-form, two B–Z junctions should be established at both ends of that sequence [47].

The 15-base-pair segment of the DNA structure (PDB code: 2ACJ) contains one B–Z junction. Eight base pairs are stabilized in the Z conformation with peptides bound to each strand of the Z-DNA segment, while six base pairs retain the B conformation. At the B–Z junction, the A–T base pair is disrupted, and each base is extruded from the double helix. The B- and Z- segments of DNA are connected through the sugar–phosphate backbone [48].

The results of molecular docking on the 2ACJ structure of DNA showed that all three complexes exhibited a significantly higher affinity for binding to the region with the Z-form of DNA (Figure 5) compared to the binding on the DNA molecule in the B-form (1BNA, Figure 5). The first binding site for all three complexes is located in this region, with binding energies of −7.5 kcal/mol (for the **C1** complex), −8.1 kcal/mol (for the **C2** complex), and −8.5 kcal/mol (for the **C3** complex). The preference for binding to the Z-DNA region is so strong that the molecular docking algorithm did not identify any binding sites for the three complexes in the B-DNA portion of the 2ACJ structure. These findings suggest that the transition of CT-DNA to the Z-form is supported by intermolecular interactions with the complexes.

The binding of two **C2** complexes per CT-DNA is a potential explanation for the increased stability observed in the resulting C2-DNA adduct. This was determined through UV–Vis absorption studies and the intrinsic binding constant Kb (refer to Table 1). Conversely, the lower values of Kb constants observed for the **C1** and **C3** complexes can be attributed to the smaller number of bound complexes per DNA (only one complex per DNA). It is evident that the length of the S-isoalkyl substituents plays a crucial role in the binding of the investigated complexes to the major groove of B-form DNA. Specifically, in the **C2** complex, the length of the substituents is sufficient to bridge the phosphate backbones of both DNA strands along the major groove (refer to Figure 6), thereby forming an adduct with a binding energy comparable to that of its binding to the minor groove (refer to Figure 5). On the other hand, the reduction in the length of S-isoalkyl substituents in the **C1** complex, or their elongation in the **C3** complex, did not result in the formation of stable adducts with binding energies similar to those observed for the minor groove.

The Z-form of DNA has a significant impact on the development of human diseases, making it a potential target for future therapeutic approaches. Recent studies have shown that Z-DNA can play a crucial role in treating tumor tissues. The survival and spread of tumor tissues depend on the extracellular matrix produced by cancer-associated fibroblasts (CAFs) and their communication with tumor cells. It is important to note that CAFs themselves are not mutated cells; rather, they create a favorable environment for mutant cells to thrive in tumorigenic tissue. When conventional treatments like radiation and chemotherapy fail, immunotherapy becomes a hopeful option. Immunotherapy involves boosting the immune system to fight against cancer cells. However, this approach is not always effective in the long-term. Tumors employ a strategy that mimics healthy cells, using a protein called adenosine deaminase acting on RNA (ADAR1) to suppress the immune response against normal cells. Inflammatory CAFs (iCAFs) play a role in creating an environment that suppresses the immune system and facilitates the spread of cancer to distant sites. If CAFs undergo processes like necroptosis, reprogramming, or lose their ability to communicate with tumor cells, it could open up potential avenues for successful tumor treatment.

Z-DNA is associated with the regulation of immune responses to viral infections. Z-DNA-binding protein 1 (ZBP1) is an important player in this response. ZBP1 possesses Zα domains that can recognize specific sequences on regular B-DNA, known as flipons, and induce the formation of Z-DNA. Once bound to the Z-DNA structure, ZBP1 influences the cells in tumor tissues, tricking the immune system into perceiving a viral infection, thus destroying the tumor tissue. This drug is effective regardless of the specific mutation causing the cancer and also eliminates fibroblasts that support tumor growth, thereby enhancing the effectiveness of immunotherapy [49].

### 2.3. Results Antitumor Activity

#### 2.3.1. Effects of Copper(II) Complexes on Tumor Murine Colon Carcinoma Cells Viability In Vitro

The previous experimental study indicates that copper(II) complexes with thiosalicylic acid exhibit considerable anticancer capacity in human colorectal cancer cells [1,24]. Therefore, the following aim was to test the cytotoxic effects of these complexes on murine colon carcinoma cells as well as their possible in vivo effects in a murine heterotopic model of primary colon carcinoma.

The MTT assay was used to evaluate the cell viability of three complexes against murine colon cancer (CT26) cells as well as human colorectal cancer cells (SW480) with cisplatin as a control. All tested copper(II) complexes showed a dose dependent cytotoxic effect toward the SW480 and CT26 cells, but was lower in comparison with cisplatin (Figure 7). Cytotoxic activity of all of the tested copper(II) complexes was higher against the murine colon carcinoma cells in comparison with the human colorectal cells. **C3** showed the highest cytotoxicity toward CT26 cells, while **C1** exhibited the lowest effect.

The morphology of CT26 cells after 24 h of exposure to copper(II) complexes and cisplatin is shown in Figure 8. A reduced number of adherent CT26 cells could be observed after treatment with **C1** at a concentration of 62.5 µM, while a smaller decrease in the number of adherent CT26 cells was observed after treatment with complexes **C2** and **C3**. The highest reduction in adherent CT26 cells was observed after treatment with cisplatin. These results are consistent with the results obtained by the MTT assay (Figure 8). **C1** and **C3** were also observed to change the morphology of CT26 cells, causing rounding of these cells, especially **C1** (Figure 8). Typical manifestations of apoptosis were not detected, nor in cell culture treated with cisplatin, nor in the cultures treated with copper(II) complexes, although some level of cell deformation characteristic of apoptosis was noticed in the culture of cells treated with **C1**.

The analysis of the IC_50_ values calculated for the tested copper(II) complexes and cisplatin (Table 3 showed that **C3** exhibited the strongest cytotoxic effect on all of the tested cell lines. Likewise, **C1** showed the weakest cytotoxic effect, followed by **C2** with better activity, and then, as mentioned, **C3** (Table 3). **C3** showed better activity than cisplatin on the mouse colon cancer cell line CT26 (Table 3).

The analysis of the selectivity index revealed that **C3** exhibited greater cytotoxicity on tumor cells than on the normal fibroblasts, with a selectivity index greater than 1 for both of the examined tumor cell lines (Table 4. Furthermore, **C3** had a higher selectivity index than cisplatin calculated for the murine colon carcinoma cell line. This finding indicates that **C3** may possibly exhibit less toxicity and organ damage than cisplatin in vivo.

#### 2.3.2. Effects of Copper(II) Complexes on Tumor Cell Apoptosis

Since we previously showed that these complexes induced moderate apoptosis of human colorectal cancer cells and significant apoptotic death of lung cancer cells [24], we used Annexin V FITC/PI double staining to measure the apoptotic rate of the copper(II) complex treated CT26 cells. Flow cytometry analysis showed that the percentages of apoptotic CT26 were higher compared with the untreated cells, but lower compared with the cisplatin treated cells (Figure 9). However, the percentage of early apoptotic CT26 cells treated with **C1** was significantly higher in comparison with the untreated cells (Figure 9). This result is in accordance with the phase contrast microscopy of CT26 cells when cell rounding was observed in the cultures treated with **C1** (Figure 8).

#### 2.3.3. Effects of **C3** on Cell Cycle in CT26 Cells

In order to investigate the inhibitory effects of copper(II) complexes on the proliferation of CT26 cells, the cell cycle distribution after a 24 h treatment was explored. The percentages of **C3**-treated CT26 cells in the G2/M phase were significantly higher than those in the control group (Figure 10). Similarly, cisplatin-induced accumulation of CT26 cells in the G2/M phase (Figure 10). **C1** and **C2** induced no cell cycle disturbances. When the extent of DNA damage is in excess to the DNA repair potential, a cell can give itself additional time at the G1/S or G2/M checkpoint to repair DNA damage [25]. If the repair falls, the cell can activate a programmed death pathway or stop cellular division, transiently (quiescence) or permanently (senescence), or can force cells into mitosis, leading to mitotic catastrophe, which results in senescence and the inhibition of the clonogenic activity of tumor cells [26]. Thus, although **C3** does not induce significant apoptosis of the CT26 cells, the highest potential to reduce the viability of CT26, as evaluated by the MTT assay, could be explained by **C3**, which induced arrest in the cells in G2/M and reduced the CT26 viability by other mechanisms [26].

#### 2.3.4. Effects of **C3** on the Growth and Metastasis of Murine Primary Heterotopic Colon Cancer

Protective effects of aspirin (acetylsalicylic acid) against colon cancer are well-known. Protective effects of acetylsalicylic acid are explained by several mechanisms such as the inhibition of cyclooxygenases, induction of apoptosis, inhibition of NF-κB activity, upregulation of tumor suppressor genes, and the inhibition of mTOR signaling [27]. These activities are probably mediated by salicylate, the main metabolite of aspirin [50]. Salicylate inhibits lysine acetyltransferases and MUC1 induces epithelial to mesenchymal transition, which could inhibit colorectal cancer progression since overexpressed MUC1 in colorectal cancer cells is associated with worse prognosis [51]. Since an inflammatory environment promotes cellular proliferation, angiogenesis, and apoptotic resistance, and salicylates exert anti-inflammatory activities in vivo, we next analyzed the antitumor effects of the **C3** complex with the thiosalicylic derivative in vivo in a heterotopic model of primary murine colon carcinoma.

Four days after receiving the last dose of the tested complex, the tumor volume in the group of mice treated with complex **C3** was significantly reduced compared to the group of untreated mice (Figure 11a). Cisplatin treatment, administered in the same manner as the copper complex treatment, also reduced the growth of the primary tumor, but the reduction in tumor volume did not reach statistical significance. Tumor weight was significantly reduced in mice treated with complex **C3** compared to the untreated animals (Figure 11b). Cisplatin treatment reduced the tumor weight compared to the untreated animals, but this difference did not reach statistical significance (Figure 11b). Moreover, the mean value of the weight of the excised primary tumor in the group of mice receiving **C3** was significantly lower compared to the mean value of the tumor weight in the group of mice treated with cisplatin (Figure 11b).

The expression of molecules involved in the process of apoptosis (mRNA level) was analyzed in the tissue of the primary tumor by real-time RT-PCR in all three groups of mice. The mRNA expression of the proapoptotic Bax and caspase 3 was significantly higher (r < 0.05) in the group of mice treated with the **C3** complex compared to the group of untreated mice (Figure 12). The expression of these molecules in the group of mice treated with cisplatin was higher compared to the group of untreated mice, but it was lower than in the group of mice treated with complex **C3** (Figure 12). The finding of the highest expression of molecules involved in apoptosis in the primary colon cancer tissue of mice treated with **C3** was consistent with the finding of the smallest tumor volume and weight in this group.

#### 2.3.5. **C3** Reduces Liver and Lung Metastases of Mice with Primary Heterotopic Colon Cancer CT26

Liver and lung tissues were analyzed 24 days after tumor cell inoculation in the groups of untreated mice that had received **C3** or cisplatin intraperitoneally. Histological analysis of five serial sections of liver and lung tissue stained with hematoxylin and eosin was performed.

Histological analysis of the lungs of the treated and untreated mice revealed that **C3** reduced the incidence of lung metastases (Table 5. Both **C3** and cisplatin reduced the percentage of mice that developed lung metastases after subcutaneous inoculation of the colon carcinoma cells. The results show that the incidence of metastasis in the group of mice that received **C3** was twice as small compared to the group of mice that received cisplatin.

The size of individual metastases was the largest in the group of untreated mice (Figure 13a,b), smaller in the group of mice treated with cisplatin (Figure 13e,f), and the smallest in the group of mice treated with **C3** (Figure 13i,j). In the group of untreated mice, metastases that affected the entire lobes of the lungs were observed (Figure 13c,d). On the other hand, in the lungs of treated mice, bleeding fields surrounding the metastases were observed (Figure 13g,h,k,l). Bleeding was more pronounced in the lungs of mice treated with cisplatin than in the group of mice treated with complex **C3**.

The findings obtained from the analysis of liver sections were similar to the findings from the analysis of lung tissue. **C3** reduced the incidence of liver metastases (Table 6). Untreated mice subcutaneously injected with CT26 cells developed liver metastases in 88.89% compared with 50% of mice with liver metastases that had received cisplatin and 25% of mice with liver metastases in the group of mice that had received **C3** (Table 6). Similar to the lungs, the areas of metastases in the livers of the untreated mice were larger (Figure 14a,b) compared to the size of the metastases in the livers of mice treated with cisplatin (Figure 14d,e). Minimal metastases, incipient, surrounding blood vessels or bile ducts were observed in the livers of mice treated with **C3** (Figure 14g,h). The number of metastases per liver was the highest in the group of untreated mice, lower in the group of mice treated with cisplatin, and the lowest in the group of mice treated with **C3**. Fields of hemorrhage were observed in the livers of all three groups of mice. The smallest area with hemorrhage were observed in the livers of mice treated with complex **C3** (Figure 14i) in comparison with the livers of the untreated mice (Figure 14c) and mice treated with cisplatin (Figure 14f).

#### 2.3.6. **C3** Reduces the Expression of Inflammation Markers in Tissue of Primary Murine Heterotopic Colon Cancer

Inflammation plays an important role in both the initiation and progression of colorectal cancer. Inflammation initiated by tumor cells promotes tumor growth and the formation of distant metastases [52]. Since the examined copper complex was conjugated with a thiosalicylic acid derivative, the potential effect of the complex in reducing inflammation in the tumor tissue was also examined. The influence of **C3** on the expression of inflammatory molecules TNF-α, pro-IL-β, ICAM-1, and VCAM-1 in the tissue of the primary tumor was examined using the real-time RT PCR method. As seen in Figure 15, **C3** significantly reduced the expression (mRNA level) of inflammatory molecules TNF-α, pro-IL-β, ICAM-1, and VCAM-1 in the tumor tissue, unlike cisplatin, which significantly reduced the expression of pro-IL-β and ICAM-1.

## 3. Discussion

Significant progress is being made in the development of drugs to combat previously incurable forms of cancer. One approach involves targeting and disrupting the structure of DNA to inhibit replication and transcription. In this study, we focused on Cu(II) complexes containing S-isoalkyl derivatives of salicylic acid (referred to as **C1**, **C2**, and **C3**) for this purpose. Through a combination of experimental and theoretical methods, we investigated how these complexes interacted with DNA.

Our experiments using ethidium bromide (EB) revealed that the complexes did not displace EB, indicating a lack of intercalation. Molecular docking studies provided insights into the binding preferences of the complexes, suggesting that they may bind in the major groove to the sugar–phosphate backbone, or, most favorably, in the minor groove. Viscosity experiments showed a modest increase in the viscosity of DNA solutions upon the addition of the **C1** or **C3** complexes, indicating some interaction. Each complex exhibited a single binding site in the minor groove. However, the viscosity strikingly increased when DNA was mixed with the **C2** complex, which was bound at two distinct binding sites—first in the minor groove, and the second position in the major groove. The presence of two bulky **C2** complexes of Cu(II) induced the transformation of B-DNA into Z-DNA. The binding energies calculated for all three Cu(II) complexes to Z-DNA were notably higher compared to B-DNA. Z-DNA had an elongated structure compared to B-DNA, which explains the significant increase in viscosity observed when DNA interacted with two **C2** complexes simultaneously.

In this study, it was shown that the binuclear complex of copper(II) with the S-isoamyl derivative of thiosalicylic acid, **C3**, arrests cancer cells in the G2/M phase, achieves a cytotoxic effect in murine colon cancer cells, and significantly reduces the expression of inflammatory cytokines in the tissue of primary CT26 mouse colon cancer, resulting in a significant reduction in primary tumor growth and metastasis.

Copper(II) complexes with alkyl derivatives of thiosalicylic acid were previously synthesized and characterized [53] and their antitumor activity was analyzed [24]. These copper complexes were shown to have a weak cytotoxic effect on the murine colon cancer cells CT26 and CT26. CL25, and human colorectal cancer cells HCT116 [24]. Palladium complexes with the same ligands showed a better antitumor effect on human colorectal cancer cells, HCT116 and CaCo-2, and human lung adenocarcinoma, A549, compared with the same copper complexes [53,54,55]. Zn(II) complexes with the same ligands showed relatively moderate cytotoxicity on murine, 4T1, and human, MDA-MB-468, breast carcinoma cells [56]. Platinum(IV) complexes with the same ligands showed significant cytotoxicity on murine leukemia cell, BCL1, and a moderate to weak effect on human B-prolymphocytic leukemia cell, JVM-13 [57]. The dinuclear complex 3 investigated in this study showed a better cytotoxic effect on murine colon carcinoma cells compared to similar mononuclear complexes [24]. In this study, the **C3** complex also showed a better cytotoxic effect toward the murine colon cancer line CT26 than on the human colorectal cancer cell line SW480, in contrast to the study showing that the Cu(II) complex with goserelin acetate exhibited better cytotoxic effect on A549 human lung adenocarcinoma cells than on the CT26 murine colon carcinoma cell line [58]. The different effect of dinuclear copper complexes with S-isoalkyl derivatives of thiosalicylic acid on different tumor cells is consistent with the results of a recent study claiming that Cu(II) complexes with derivatives of Schiff’s bases exert different effects on colorectal cancer cells DLD-1 and human breast cancer cells MDA-MB-231 [59]. The in vitro cytotoxicity of the copper complexes examined in this study was weaker compared to the similarly examined cytotoxicity of the Cu(II) complexes with pyrrolysines [60]. Similar to the results of this study, a moderate cytotoxic effect of Cu(II) complexes with tridentate Schiff bases with different halogens was demonstrated on HCT116 human colorectal cancer cells, while the same complexes showed a very pronounced effect on human ovarian cancer cells [61]. Very good cytotoxic effects of Cu(II) complexes with salicylic acid derivatives on HT29 human colorectal cancer cells were described [62].

Compounds exhibiting antitumor activity often achieve this activity by interacting with different molecules such as DNA, proteins, and parts of the cell membrane [63]. The dinuclear copper complexes examined in this study had relatively weak interactions with human serum albumin and the DNA molecule, and the strongest interactions with these two molecules were achieved by complex **C2** (Table 1 and Table 2). As the best cytotoxicity in vitro was achieved by complex **C3**, it is most likely that interaction with DNA contributes little to the antitumor activity of dinuclear Cu(II) complexes with S-isoalkyl derivatives of thiosalicylic acid.

Copper(II) complexes with alkyl derivatives of thiosalicylic acid exert a moderate apoptotic effect on murine and human colorectal cancer cells [24]. In accordance with that finding, there was a relatively weak apoptotic effect of dinuclear complexes of copper(II) with S-isoalkyl derivatives of thiosalicylic acid on CT26 cells (Figure 9). Good apoptotic activity of various Cu(II) on human colorectal cancer cell lines has been previously described [64,65,66,67]. Similar to the results of this study, Cu(I) thiocyanate complexes with phosphine derivatives of sparfloxacin exert a mild to moderate apoptotic effect on the CT26 cells [68]. Chromosome instability, which precedes in 60–80% of colorectal cancers, very often affects genes whose products participate in the control of apoptosis. The p53 gene is mutated in about 70% of colorectal cancers [69]. The p53 protein participates in the regulation of the cell cycle, DNA repair mechanisms, and apoptosis; which of these processes is the result of p53 activation, depends on the cell type and microenvironment [70]. The p53 mutation in HCT116 human colorectal cancer cells makes these cells more sensitive to radiotherapy and adriamycin, and less sensitive to 5-fluorouracil [71]. It is possible that the mutations in CT26 cells make these cells relatively resistant to apoptosis induced by dinuclear complexes of copper(II) with alkyl derivatives of thiosalicylic acid, which does not exclude other mechanisms of antitumor activity.

Cells undergoing mitotic catastrophe are in the G2/M phase of the cell cycle [72]. Mitotic catastrophe is a disrupted mitosis resulting in cell death, which differs in mechanism from apoptotic or necrotic death. The **C3** complex, which exhibited the strongest cytotoxic activity on all examined tumor cell lines (Table 3), also caused the accumulation of CT26 cells in the G2/M phase of the cell cycle (Figure 10). Thus, although **C3** induced no significant apoptosis of CT26 cells, its greatest potential to reduce CT26 viability, assessed by the MTT assay, could be explained by the arrest of CT26 cells in the G2/M phase of the cell cycle under the action of **C3** and the reduction in CT26 viability by other mechanisms (such as mitotic catastrophe) [73]. A similar effect of the induction of mitotic catastrophe and subsequent cell death of human non-small cell lung carcinoma, H-460, was described for the ternary complex of Cu(II) with L-tyrosine and diimines [74].

Numerous studies have shown that aspirin (acetylsalicylic acid) reduced the rate of colorectal adenoma recurrence, advanced adenoma, the number of recurrent adenomas, and delayed the time to adenoma recurrence [75,76,77,78]. It also reduced the risk of developing colorectal cancer [79,80]. Aspirin acts as an inhibitor of cyclooxygenase-2. It reduces the production of PGE2 and reduces inflammation, therefore reducing the risk of the development and progression of colorectal cancer [81]. Thiosalicylic acid, like aspirin, belongs to the group of non-steroidal anti-inflammatory drugs, so it is possible that metal complexes with derivatives of this acid such as the dinuclear Cu(II) complex with the S-isoamyl derivative of thiosalicylic acid achieve an enhanced antitumor effect in vivo in colon cancer by acting directly on tumor cells, but also by acting indirectly by inhibiting inflammation. **C3** significantly reduced the growth of the primary heterotopic CT26 mouse carcinoma compared to untreated mice (Figure 11). An even more significant result is the significantly lower weight of the excised primary tumor in the group of mice treated with complex **C3** compared to the group of mice treated with cisplatin (Figure 11). The treatment of mice with the **C3** complex was accompanied by a lower incidence of metastases (Table 5) in the lungs and liver and a smaller size of metastatic changes (Figure 13 and Figure 14), which is in accordance with the strongest cytotoxic activity of **C3** on CT26 cells, as evaluated by MTT. In the tissue of the primary tumor of mice treated with the **C3** complex, a significantly higher expression of mRNA molecules that are crucial for the process of apoptosis, Bax and caspase-3, was recorded (Figure 12), and a significantly lower expression of the proinflammatory molecules TNF-α, pro-IL-β, and ICAM-1 and VCAM-1 (Figure 15) were detected compared to the tissue from untreated mice.

Blocking IL-1β in a murine model of colitis and associated colon cancer significantly reduces the size and invasiveness of tumors [82,83], which indicates a significant role of this cytokine in the development and progression of colorectal cancer. TNF-α promotes the migration and invasiveness of colorectal cancer [84] and participates in colorectal carcinogenesis associated with ulcerative colitis [85]. VCAM-1 is a molecule that is expressed in colorectal cancer; it is associated with the presence of CD3+ lymphocytes [86] and participates in tumor infiltration by immune cells. ICAM-1 is a molecule whose expression is controlled by the cytokines TNF-α and IL-β, which plays an important role in colorectal pathogenesis, participating in mutual interactions between the tumor and extracellular matrix in signal transduction and numerous immune processes [87]. Colorectal cancer cells, in contrast to normal intestinal epithelial cells, express ICAM-1, which, through interaction with ligands, promotes tumor metastasis [88,89]. The expression of ICAM-1 determines the malignant potential of colorectal cancer; it is correlated with a worse prognosis [90,91], although there are also studies indicating that the membrane expression of this molecule correlates with better prognosis of colorectal cancer [92], which is explained by the possibility that ICAM-1 expression on the membrane of tumor cells stimulates the killing of tumor cells by cytotoxic lymphocytes, if they are present in the tumor microenvironment. Soluble ICAM-1 promotes angiogenesis and the growth of colorectal cancer [93]. Greater expression of ICAM-1 molecules in tumor tissue can also result in enhanced tumor cell apoptosis [94].

The binuclear copper(II) complex with the S-isoamyl derivative of thiosalicylic acid achieved a very convincing antitumor effect in vivo in a mouse colon cancer model, significantly reducing the growth of the primary tumor and the incidence and size of metastases in the lungs and liver. As the in vitro antitumor effect of this complex is moderate, it is possible that the reduced growth and metastasis of murine colon cancer is a consequence of the modulation of the tumor microenvironment by the tested complex. Bearing in mind the importance of inflammation for the formation and progression of colorectal cancer [81,82,83,84,85], it is possible that **C3**, by influencing the expression of inflammatory molecules TNF-α, pro-IL-β, ICAM-1, and VCAM-1 in the tissue of the primary tumor, reduces tumor growth and progression, and also stimulates the apoptosis of tumor cells in vivo.

## 4. Materials and Methods

### 4.1. Materials and Measurements

All reagents were obtained commercially and used without further purification. The kinetics and mechanism of the substitution reactions of the complexes with selected ligands were studied in a Stopped-flow spectrophotometer in 25 mM Hepes buffer, pH ≈ 7.2 (Acros Organics, Geel, Belgium).

### 4.2. Syntheses

#### 4.2.1. General Procedure for the Synthesis of S-Isoalkyl Derivatives of Thiosalicylic Acid (**L1**)–(**L3**)

The S-isoalkyl derivatives of thiosalicylic acid ligands (isoalkyl = isopropyl- (**L1**), isobutyl- (**L2**), isopentyl- (**L3**)) were prepared by the alkylation of thiosalicylic acid by means of the corresponding alkyl halides in alkaline water–ethanol solution [24].

#### 4.2.2. General Procedure for the Synthesis of Copper(II)-Complexes with S-Isoalkyl Derivatives of Thiosalicylic Acid (**C1**)–(**C3**)

The binuclear copper(II) complexes with S-isoalkyl derivatives of thiosalicylic acid as ligands (isoalkyl = isopropyl- (**C1**), isobutyl- (**C2**), isopentyl- (**C3**)) were prepared by the direct reaction of copper(II)-nitrate and corresponding S-isoalkyl derivatives of thiosalicylic acid with the addition of a water solution of LiOH in a molar ratio of 1:2:2 [24].

### 4.3. UV–Vis DNA Interactions

CT-DNA stock solutions were prepared in PBS, resulting in a UV absorbance ratio A_260_/A_280_ of ca. 1.8–1.9, indicating negligible protein contamination. CT-DNA concentrations were determined using A_260_ with ε = 6600 M^−1^cm^−1^ [95]. Fluorescence spectra were recorded in the range 550–750 nm with excitation at 527 nm. Excitation and emission bandwidths were both 10 nm.

### 4.4. UV–Vis Absorption Studies

To quantitatively compare the binding strength of the complexes, the intrinsic binding constant *K*_b_ was determined by monitoring the changes in absorption at the MLCT band with an increasing concentration of CT-DNA using Equation (2).
[DNA]/(*ε*_A_ − *ε*_f_) = [DNA]/(*ε*_b_ − *ε*_f_) + 1/[*K*_b_(*ε*_b_ – *ε*_f_)](2)

*K*_b_ is given by the ratio of the slope to the y intercept in plots of [DNA]/(*ε*_A_ − *ε*_f_) vs. [DNA], where [DNA] is the concentration of DNA in base pairs, *ε*_A_ = A_obsd_/[complex], ε_f_ is the extinction coefficient for the unbound complex, and *ε*_b_ is the extinction coefficient for the complex in the fully bound form.

### 4.5. Ethidium Bromide (EB) Displacement Studies

The relative binding of complexes to CT-DNA was determined by calculating the quenching constant (*K*_sv_) from the slopes of straight lines obtained from the Stern–Volmer equation (Equation (3)).
I_0_/I = 1 + *K*_sv_[Q](3)
where I_0_ and I are emission intensities in the absence and presence of the quencher (complexes **C1**, **C2**, and **C3**), respectively; [Q] is the total concentration of the quencher; *K*_sv_ is the Stern–Volmer quenching constant obtained from the slope of the plot of I_0_/I vs. [Q].

### 4.6. Viscosity Measurements

The viscosity of the aqueous DNA solution was measured in the presence of increasing amounts of complexes **C1**, **C2**, and **C3** using an Ubbelohde viscosimeter (SI Analytics GmbH, Mainz, Germany, type no. 525 03) by measuring the flow rate. The viscosimeter was filled with experimental liquid and placed vertically in a glass-sided thermostat maintained constantly at ±0.01 K, with a standard uncertainty of controlled temperature of ±0.02 K. Flow time was recorded with a digital stopwatch with an accuracy of ±0.001 s, afterward, thermal equilibrium was obtained. All measurements were performed at 310.0 K. The data are presented as (η/η_0_)^1/3^ against r, where η is the DNA viscosity in the presence of complex and η_0_ is the viscosity of DNA in the buffer alone. Viscosity values were calculated from the observed flow time of DNA-containing solutions (t) corrected for the flow time of buffer alone (t_0_),
η = (t − t_0_)/t_0._

### 4.7. Protein Binding Studies

Protein fluorescence is due to natural fluorophores such as tryptophan, tyrosine, and phenylalanine. Changes in BSA fluorescence were used to monitor the interaction with metal complexes. Tryptophan fluorescence quenching experiments were conducted using 2.0 μM BSA in PBS. Quenching of the emission intensity of BSA tryptophan residues at 363 nm in the presence of increasing concentrations of Cu(II) complexes **C1**, **C2**, and **C3** (0–20 µM) was monitored. Fluorescence spectra were recorded in the range 300–500 nm with excitation at 295 nm. The excitation and emission bandwidths were both 10 nm.

### 4.8. Methodology of Docking

The structures of the Cu(II) complexes (**C1**, **C2**, and **C3**) and ethidium bromide (EB) were used as ligands in the docking studies. The structures of the Cu(II) complexes differ based on the alkyl chain length of the substituents on aromatic ligands coordinated to Cu^2+^ ions. Complexes labeled with higher numbers have longer alkyl groups. The optimization and calculation of Merz–Kollman atomic charges for the Cu(II) complex structures were performed by the B3LYP method using a 6-31G** basis set. The crystal structure of ethidium bromide was optimized in ArgusLab using the Molecular Mechanics Universal Force Field. Cu(II) complexes are neutral structures, while the positively charged form of ethidium bromide was used in all calculations.

The receptors utilized in the study consisted of DNA structures extracted from Protein Data Bank (PDB). These DNA structures underwent a cleaning process that involved the removal of solvent molecules and ligands. One of the DNA structures corresponded to a regular double strand in the B-form, specifically identified as PDB entry 1BNA. The second DNA structure (PDB code: 2ACJ) exhibited a combination of both B-form and Z-form conformations [96].

The Br^-^ ion was removed from the ethidium bromide structure file. In reality, the ethidium cation is bound to a DNA molecule. Nevertheless, in this study, due to simplicity, the ethidium cation was labeled as EB. AutoDockTools4 software, Version 1.5.7, was used for the preparation of the receptors and metal complexes, and the determination of docking parameters [97]. Two groups of docking studies were performed with the program AutoDockVina [98]. In all of the docking studies, ligands were docked on the whole DNA structure. Visualization and analysis were completed in BIOVIA Discovery Studio [99]. In order to estimate the influence of EB on the docking of complexes on the intercalation site, the structure of the DNA with intercalated EB was used as a receptor for the complexes targeting the intercalation site occupied by EB.

### 4.9. Cytotoxic Activity

#### 4.9.1. Cell Culture

Murine colon cancer (CT26) and human colorectal cancer (SW480) cell lines as well as cell lines of murine fibroblasts (3T3) and human fibroblasts (MRC5) were purchased from American Type Culture Collection. Cells were routinely grown in complete DMEM (Sigma Aldrich) in a 5% CO_2_ incubator with standard conditions. In all experiments, only cell suspensions with >95% viable cells were used. Trypan blue was used to determine the number of viable tumor cells.

#### 4.9.2. MTT Assay

The cytotoxic effects of the three copper(II) complexes on the CT26 cells were determined using the MTT colorimetric technique. Cells were seeded on 96-well plates at a density of 5·10^3^ cells/100 μL (per well) in complete DMEM growth medium and allowed to adhere by incubation at 37 °C overnight. After 24 h, the culture medium was replaced and each well received 100 μL of different compounds, which had been serially diluted two-fold in the medium to concentrations ranging from 1000 to 7.8 μM. In the presence of copper complexes, cells were incubated under standard conditions (37 °C/5% CO_2_) for 72 h. Upon incubation, the medium was removed, the MTT (3-(4,5-dimethylthiazol-2-yl)-2,5-diphenyltetrazolium bromide) solution (5 mg/mL in PBS, 20 μL) was added to each well and the 96-well multiplates were incubated for an additional 4 h. Using a microplate multimode detector Zenyth 3100(Anthos Labtec Instruments GmbH, Wals, Austria) the optical density of each well was determined at 595 nm. The percentage of cell viability was determined by a comparison with the untreated controls according to formula: % of viable cells = (E − B)/(S − B) × 100, where B stands for the background of medium alone; S is the total viability/spontaneous death of untreated target cells; E is the experimental well. Experiments, performed in triplicates, were repeated three times.

#### 4.9.3. Annexin V Propidium Iodide Double Staining Assay

After 24 h of treatment with the tested compound at a concentration of 62.5 μM, CT26, the cells were collected, washed in PBS, and resuspended in ice cold binding buffer [10× binding buffer: 0.1 M Hepes/NaOH (pH 7.4), 1.4 M NaCl, 25 mM CaCl_2_]. AnnexinV-FITC (BD Pharmingen, San Diego, CA, USA) and propidium iodide (Sigma Aldrich, Darmstadt, Germany) were added to each sample and incubated in the dark for 15 min. The percentage of dead cells was determined by a FACS Calibur flow cytometer (BD Biosciences, San Jose, CA, USA) and the data were analyzed using FlowJo (Tree Star).

#### 4.9.4. Cell Cycle Analysis

CT26 cells were allowed to grow until a confluency of 70–80% was achieved in the culture plates. They were exposed to copper(II) complexes or cisplatin (62.5 μM) for 12 h and the cell cycle analysis was performed with Vybrant^®^ DyeCycle™ Ruby stain (Invitrogen Molecular probes, Eugene, Oregon, USA) according to the manufacturer’s instructions. After the treatment, CT26 cells were stained with Vybrant DyeCycle Ruby and analyzed by a FACS Calibur flow cytometer (BD Biosciences, San Jose, CA, USA). The cell cycle distribution was analyzed using FlowJo software, Version 7.6.5 

#### 4.9.5. Experimental Animals

All experiments were approved by and conducted in accordance with the Guidelines of the Animal Ethics Committee of the Faculty of Medical Sciences of the University of Kragujevac, Serbia. Eight to ten week old BALB/c mice were used. The mice were housed in a temperature-controlled environment with a 12-h light-dark cycle, fed ad libitum, and observed daily. Experimental animals were also equalized in weight and randomized in the experimental or control groups.

#### 4.9.6. Heterotopic Model of Murine Colon Cancer and Drug Treatment

BALB/c mice were inoculated with 1 × 10^6^ CT26 cells subcutaneously at the back. Seven days after the CT26 cells were applied, the mice were injected with the complex or cisplatin by intraperitoneal injection. Mice received, from the eighth day after tumor cell inoculation, either cisplatin or the **C3** complex (5 mg/kg body weight/for 5 consecutive days followed by 2 days break and treated again for 5 days), or phosphate-buffered saline. Mice were sacrificed on the twenty-fourth day of the experiment.

#### 4.9.7. Estimation of Colon Cancer Growth

The size of the primary heterotopic CT26 tumors was assessed morphometrically in two dimensions by caliper. The tumor volumes (mm^3^) were calculated according to the formula: tumor volume (mm^3^) = L (major axis of the tumor) × W(minor axis)^2^/2.

#### 4.9.8. Histological Analysis

The tissue sections of the livers and lungs were fixed in 4% paraformaldehyde and embedded in paraffin, cut into thin sections, mounted on glass slides, and stained with hematoxylin and eosin. The slides were evaluated for the presence of metastases under low-power light microscopy (BX51; Olympus, Tokyo, Japan) equipped with a digital camera.

#### 4.9.9. RNA Extraction and Real-Time qRT-PCR

Total RNA from the tissue of primary tumors was extracted using TRIzol (Invitrogen, Carlsbad, CA, USA). Total RNA (2 μg) was reverse-transcribed to cDNA using the RevertAid H Minus First Strand cDNA Synthesis Kit (Thermo Fisher Scientific, Vilnius, Lithuania). qRT-PCR was performed using Luminaris Color HiGreen qPCR Master Mix (Thermo Fisher Scientific) and miRNA specific primers presented in Table 6 in a Mastercycler ep realplex (Eppendorf, Hamburg, Germany). The relative expression of genes was calculated according to the formula 2−(Ct−Ct actin), where Ct is the cycle threshold of the gene of interest and Ct actin is the cycle threshold value of the housekeeping gene (GAPDH).

#### 4.9.10. Statistics

Statistical analyses were performed using SPSS Version 20.0 for Windows software (SPSS Inc., Chicago, IL, USA). The results were analyzed using the Student’s *t*-test. The difference was considered significant when *p* < 0.05.

## 5. Conclusions

The binuclear Cu(II) complexes investigated in this study demonstrated their ability to impede the functioning of DNA by specifically binding to its minor groove. Among these complexes, the **C2** complex is particularly intriguing due to its unique property of inducing the formation of Z-DNA. Z-DNA is a distinct DNA conformation that has been recognized for its potential to augment the immune response against tumor tissues. Furthermore, these copper(II) complexes, especially **C3**, arrest tumor cells in the G2/M phase of the cell cycle and significantly reduce the viability of murine colon cancer cells. **C3** also reduces the expression of inflammatory molecules in primary tumor tissue in a mouse model of heterotopic colon carcinoma.

These findings suggest that the Cu(II) complexes, especially **C2** and **C3**, have the potential to serve as promising candidates for targeted cancer therapy. Further research and optimization of these complexes could pave the way for effective treatments against various forms of cancer.

## Data Availability

The data presented in this study are available on reasonable request from the corresponding author.

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
