# Peer review of "Docking Studies, Cytotoxicity Evaluation and Interactions of Binuclear Copper(II) Complexes with S-Isoalkyl Derivatives of Thiosalicylic Acid with Some Relevant Biomolecules"

_ijms, 2023, doi:10.3390/ijms241512504_

Round 1
Reviewer 1 Report
Manuscript ID: ijms-2511015
The Article "Docking studies, cytotoxicity evaluation and interactions of binuclear copper (II)-complexes with S-isoalkyl derivatives of thiosalicylic acid with some relevant biomolecules" by Jelena D. Dimitrijević et al submitted to International Journal of Molecular Sciences
Authors have disclosed their recent results on the interactions of earlier reported copper (II)-complexes containing S-isoalkyl derivatives the salicylic acid with guanosine-5′-monophosphate and calf thymus DNA (CT-DNA) and their antitumor effects in colon carcinoma model. They have provided Molecular docking studies which show significant affinity of the complexes for binding sites.
This manuscript is well documented and provides analytical supporting information with proper citation.
However, authors should consider providing details regarding compounds that have mentioned (L1, L2, L3, C1, C2, C3) including supporting information / analytical data
Author Response
The Article "Docking studies, cytotoxicity evaluation and interactions of binuclear copper (II)-complexes with S-isoalkyl derivatives of thiosalicylic acid with some relevant biomolecules" by Jelena D. Dimitrijević et al submitted to International Journal of Molecular Sciences
Authors have disclosed their recent results on the interactions of earlier reported copper (II)-complexes containing S-isoalkyl derivatives the salicylic acid with guanosine-5′-monophosphate and calf thymus DNA (CT-DNA) and their antitumor effects in colon carcinoma model. They have provided Molecular docking studies which show significant affinity of the complexes for binding sites.
This manuscript is well documented and provides analytical supporting information with proper citation.
However, authors should consider providing details regarding compounds that have mentioned (L1, L2, L3, C1, C2, C3) including supporting information / analytical data.
answer
We accepted suggestions and added the information regarding complexes and ligands, we added schematic drawing of their structure in Scheme 1.
L1 = S-isopropyl derivatives of thiosalicylic acid (S- isopropyl-thiosal)
L2 = S-isobutenyl derivatives of thiosalicylic acid (S-isobutenyl-thiosal)
L3 = S-isopentyl derivatives of thiosalicylic acid (S- isopentyl-thiosal)
C1 = copper(II)-complexes with S-isopropyl derivatives of thiosalicylic acid [Cu2(S-isopropyl-
-thiosal)4(H2O)2]
C2 = copper(II)-complexes with S-isobutenyl derivatives of thiosalicylic acid [Cu2(S-isobutenyl
-thiosal)4(H2O)2]
C3 = copper(II)-complexes with S-isopentyl derivatives of thiosalicylic acid [Cu2(S-isopentyl-
-thiosal)4(H2O)2]
Reviewer 2 Report
This work is aimed at characterization of the cytostatic effect of copper complexes with thiosalicylic acid derivatives and determination of possible mechanisms of interaction of these complexes with DNA molecules and proteins. The relevance of the creation of cytostatic preparations is not in doubt. The work is well structured. The results of the obtained studies are presented in the text in a very detailed and qualitative way and are a logical continuation of the previous studies of the authors. There are a small number of comments to the work:
1) Figure 7. The concentration scales on all the sub-pictures need to be unified. Also, there is an overload of this figure with text and abbreviations, so it is recommended for the authors to mark these figures with alphabetic symbols (a, b, c, d) with appropriate explanations in the caption.
2) It is necessary to correct names of subsections, including 2.3.4. and 2.3.3.
3) Some figure captions (e.g., Figure 3) give the impression of a description of the results or a detailed description of the experiment being conducted. Authors should take the experimental details out of the figure captions into the experimental part. The caption to the figures should be concise and reflect the main measured effects.
4) The authors should also explain the combination of the weak apoptotic effect of copper complexes (including C3), described in Chapter 2.3.2, with the highest stimulating effect on pro-apoptotic molecule production, described in Chapter 2.3.4 and Figure 12.
5) It is necessary to unify the design of all figures (size, fonts, resolution). Some of them are not presentable enough.
6) Section 3.6. It is required to specify the method of measuring the viscosity of complex solutions, as well as the measurement technique and the type of equipment used.
Need to be checked by a native speaker
Author Response
This work is aimed at characterization of the cytostatic effect of copper complexes with thiosalicylic acid derivatives and determination of possible mechanisms of interaction of these complexes with DNA molecules and proteins. The relevance of the creation of cytostatic preparations is not in doubt. The work is well structured. The results of the obtained studies are presented in the text in a very detailed and qualitative way and are a logical continuation of the previous studies of the authors. There are a small number of comments to the work:
- Figure 7. The concentration scales on all the sub-pictures need to be unified. Also, there is an overload of this figure with text and abbreviations, so it is recommended for the authors to mark these figures with alphabetic symbols (a, b, c, d) with appropriate explanations in the caption.
We corrected the scale, and also we presented the results as bars in the revised manuscript, we tried to reduce the text in the figure and added letter designations with explanations in figure legend.
2) It is necessary to correct names of subsections, including 2.3.4. and 2.3.3.
It is corrected and marked in revised manuscript
3) Some figure captions (e.g., Figure 3) give the impression of a description of the results or a detailed description of the experiment being conducted. Authors should take the experimental details out of the figure captions into the experimental part. The caption to the figures should be concise and reflect the main measured effects.
It has been corrected as suggested.
4) The authors should also explain the combination of the weak apoptotic effect of copper complexes (including C3), described in Chapter 2.3.2, with the highest stimulating effect on pro-apoptotic molecule production, described in Chapter 2.3.4 and Figure 12.
C3 has weak apoptotic effects in vitro (Fig 9 and chapter 2.3.2), but higher apoptotic effects in vivo (evaluated by expression of mRNA for apoptotic molecules in tissue of primary tumors) and presented in Fig 12 and chapter 2.3.4 The possible explanation is added in discussion.
The binuclear copper(II) complex with the S-isoamyl derivative of thiosalicylic acid achieves a very convincing antitumor effect in vivo in a mouse colon cancer model, significantly reducing the growth of the primary tumor and the incidence and size of metastases in the lungs and liver. As the in vitro antitumor effect of this complex is moderate, it is possible that the reduced growth and metastasis of colon cancer is a consequence of the modulation of the tumor microenvironment by the tested complex. Bearing in mind the importance of inflammation for the formation and progression of colorectal cancer, it is possible that C3, by influencing the expression of inflammatory molecules TNF-α, pro-IL-β, ICAM-1 and VCAM-1 in the tissue of the primary tumor, reduces tumor growth and progression, and also stimulate apoptosis of tumor cells in vivo.
5) It is necessary to unify the design of all figures (size, fonts, resolution). Some of them are not presentable enough.
It has been corrected as suggested.
6) Section 3.6. It is required to specify the method of measuring the viscosity of complex solutions, as well as the measurement technique and the type of equipment used.
In section 3.6. (Viscosity measurements) we have explained in detail method of measuring the viscosity of complex solutions.
Changes in DNA viscosity were measured in the presence of increasing amounts of complexes C1, C2 and C3. Flow time was measured with a digital stopwatch. Each sample was measured in triplicate and the average flow time was calculated. Data are presented as (η/η0)1/3 against r, where η is the DNA viscosity in the presence of complex and η0 is the viscosity of DNA in buffer alone. Viscosity values were calculated from the observed flow time of DNA-containing solutions (t) corrected for the flow time of buffer alone (t0), η = (t − t0)/t0.
Reviewer 3 Report
The manuscript submitted by Dimitrijevic at al represents complex interdisciplinary study and merits to be published in IJMS. Overall, the study is properly designed and the results are well discussed. However, the careful revision should be performed before the manuscript acceptance. The comments, questions and suggestions addressed for revision can be found below.
1. Introduction. Why did the authors select as isoalkyl moiety two saturated (isopropyl and isopentyl) ones and one unsaturated (isobutenyl) rather than all saturated moieties that differ in radical length? Some explanation is required.
2. Introduction/Results. A figure illustrating the chemical structures of the compounds under study would be very helpful for the readers.
3. Figures 1, 2 and 4 are incomprehensible because of poor organization and poor figure caption. In particular, the figures in the upper and lower layers are labeled with the same letters (A-C). It is better to label the lower layer with other letters and correct the caption.
4. Figure 3. Please rebuild the graph from 0 on the Y-axis. Also, instead of connecting the experimental points, it is better to use the linear fit function for the data set.
5. Table 2. What is Ksv? Please explain in the text.
6. Figure 7 is absolutely unclear. What are L1, L2 and L3 in the plots in the right column? Why are the concentrations in the left plots from 7.8 to 1000 µM the concentrations in the right column from 1000 to 7.8 µM? What is the difference between left and right plots for the same cells. The concentration values should be presented using a similar style such as 7.8 and 15.6, but not 7.8 and 15.625 μM. Please, provide the letters to each plot and give clear explanation to it in the caption. Also, add the error bars to each experimental point on the graphs.
7. Why did the authors use cisplatin (Pt complex) for cytotoxicity test but not Cu compound?
8. Figure 8. Please provide the magnification and scale bar for the images presented.
9. Captions to Figure 9-12 and 15. It should start with an indication of what is represented in the figure, and then a brief explanation of the results can be given. Each element of a complex figure should have a letter designation, which should be clarified in the figure captions.
10. Figure 13. Please provide the magnification and scale bar for the images presented. Each image should have a letter designation and reference to each subfigure should be provided in the text to better understand what results are supported by which subfigure. The key results for observation should be marked as in Figure 14.
11. Figure 14. Please provide the magnification and scale bar for the images presented.
12. The authors should explain, why C2 binds strongly to DNA than C1 and C3 basing on the differences in chemical structures of the compounds.
Some typos need to be corrected.
Author Response
The manuscript submitted by Dimitrijevic at al represents complex interdisciplinary study and merits to be published in IJMS. Overall, the study is properly designed and the results are well discussed. However, the careful revision should be performed before the manuscript acceptance. The comments, questions and suggestions addressed for revision can be found below.
- Introduction. Why did the authors select as isoalkyl moiety two saturated (isopropyl and isopentyl) ones and one unsaturated (isobutenyl) rather than all saturated moieties that differ in radical length? Some explanation is required.
First, we decided on longer alkyl/alkenyl chains because it has been confirmed that they affect the additional stabilization of binuclear and polynuclear complexes during their synthesis. When choosing the ligands for the synthesis of the complex, we opted for an isoalkenyl derivative of thiosalicylic acid (S-isobutenyl derivative of thiosalicylic acid), because we wanted to check whether the interaction with CT-DNA molecules would be similar to the case of isoalkyl derivatives. We expected the complex with S-isoalkenyl derivative of thiosalicylic acid to interact with CT-DNA in a similar way, however complex C2 is particularly intriguing due to its unique property of inducing Z-DNA formation, while copper(II)-complexes with S-isoalkyl derivatives of thiosalicylic acid did not show this. Z-DNA is a special DNA conformation recognized for its potential to enhance the immune response against tumor tissues. Through this research, we determined that the antitumor potential of the synthesized complexes primarily depends on the number of C atoms in the substituted residue of thiosalicylic acid, so that the complex with the longest alkyl chain (C3-copper(II)-complexes with S-isopentyl derivatives of thiosalicylic acid) expectedly showed the highest cytotoxic potential.
- Introduction/Results. A figure illustrating the chemical structures of the compounds under study would be very helpful for the readers.
We accepted suggestions and added the chemical structures of the compounds as the Scheme 1 in revised manuscript.
- Figures 1, 2 and 4 are incomprehensible because of poor organization and poor figure caption. In particular, the figures in the upper and lower layers are labeled with the same letters (A-C). It is better to label the lower layer with other letters and correct the caption.
We accepted suggestion and made corrections in the manuscript.
- Figure 3. Please rebuild the graph from 0 on the Y-axis. Also, instead of connecting the experimental points, it is better to use the linear fit function for the data set.
We accepted suggestion and made corrections in the manuscript.
- Table 2. What is Ksv? Please explain in the text.
The relative binding of the complex to CT-DNA was determined by calculating the quenching constant (Ksv) from the slope of the lines obtained from the Stern-Volmer equation.
I0/I = 1+ Ksv[Q]
Where I0 and I are the emission intensities in the absence and presence of quenching (for complexes C1/C2/C3), [Q] is the total concentration of quenching (DNA), and Ksv is the
Stern-Volmer constant obtained. with graph of dependence I0/I in function [Q].
- Figure 7 is absolutely unclear. What are L1, L2 and L3 in the plots in the right column? Why are the concentrations in the left plots from 7.8 to 1000 µM the concentrations in the right column from 1000 to 7.8 µM? What is the difference between left and right plots for the same cells. The concentration values should be presented using a similar style such as 7.8 and 15.6, but not 7.8 and 15.625 μM. Please, provide the letters to each plot and give clear explanation to it in the caption. Also, add the error bars to each experimental point on the graphs.
We corrected the scale, and also, we presented the results as bars +SD in the revised manuscript. We added explanation for L1, L2, and L3 in the introduction (Scheme 1). The right columns are cytotoxicity of complexes and the left columns are cytotoxicity of ligands. In revised manuscript every plot is marked with letter and the explanation is in the figure legend.
L1, L2 and L3 are abbreviations of names of ligands (derivatives of thiosalicylic acid)
L1 = S-isopropyl derivatives of thiosalicylic acid (S- isopropyl-thiosal)
L2 = S-isobutenyl derivatives of thiosalicylic acid (S-isobutenyl-thiosal)
L3 = S-isopentyl derivatives of thiosalicylic acid (S- isopentyl-thiosal)
- Why did the authors use cisplatin (Pt complex) for cytotoxicity test but not Cu compound?
Cisplatin is still used as the gold standard when monitoring the antitumor activity of complex compounds, because this complex compound first entered in clinical use. After the global success of cisplatin (cis-diammindichloridoplatinum(II)-complex) in cancer chemotherapy, the research and discovery of new complex compounds with various transition metal ions as new antitumor agents continued. This was primarily caused by the unfavorable pharmacological profile of cisplatin (non-selective effect, wide range of side effects, emergence of resistance, etc.). Research in this area is directed towards the examination of biologically active substances that will not bind covalently to the DNA chain, but will show the ability to achieve a non-covalent bond.
- Figure 8. Please provide the magnification and scale bar for the images presented.
We added the magnification in Figure legends.
- Captions to Figure 9-12 and 15. It should start with an indication of what is represented in the figure, and then a brief explanation of the results can be given. Each element of a complex figure should have a letter designation, which should be clarified in the figure captions.
We corrected the captions for figures 9-12 and 15, and added letter designation for each element of the figure with explanation in fig legend.
- Figure 13. Please provide the magnification and scale bar for the images presented. Each image should have a letter designation and reference to each subfigure should be provided in the text to better understand what results are supported by which subfigure. The key results for observation should be marked as in Figure 14.
We added magnification in figure legend, the key results are marked with arrows, letter designation for each image in the figure and reference in the main text is added.
- Figure 14. Please provide the magnification and scale bar for the images presented.
The magnification for the images is added. We do not have the possibility to add scale bar in the image with our equipment for the microscope.
- The authors should explain, why C2 binds strongly to DNA than C1 and C3 basing on the differences in chemical structures of the compounds.
Compounds exhibiting antitumor activity often achieve this activity by interacting with different molecules such as DNA, proteins, and parts of the cell membrane. In this study complex C2 (copper(II)-complexes with S-isobutenyl derivatives of thiosalicylic acid) has the strongest interactions with human serum albumin and the DNA molecule compared to other investigated compounds. That high binding nature of the metal complex can be caused by the additional π-π* interaction of the substituted residue in the ligand molecule, which is not the case with the other two complexes (C1/C3).
Round 2
Reviewer 1 Report
The Article "Docking studies, cytotoxicity evaluation and interactions of binuclear copper (II)-complexes with S-isoalkyl derivatives of thiosalicylic acid with some relevant biomolecules" by Jelena D. Dimitrijević et al submitted to International Journal of Molecular Sciences
This is a revised version of the manuscript; authors have modified manuscript as per recommendations from reviewers.
Reviewer 3 Report
The authors revised their manuscript according to the reviewer's comments. The reviewer's questions have been answered appropriately. The manuscript can be accepted in its present form.